# Double-Matrix Decomposition Image Steganography Scheme Based on Wavelet Transform with Multi-Region Coverage

**DOI:** 10.3390/e24020246

**Published:** 2022-02-07

**Authors:** Ping Pan, Zeming Wu, Chen Yang, Bing Zhao

**Affiliations:** Electronic Engineering College, Heilongjiang University, Harbin 150080, China; 2201647@s.hlju.edu.cn (P.P.); 2201669@s.hlju.edu.cn (Z.W.); 2201658@s.hlju.edu.cn (C.Y.)

**Keywords:** image steganography, multi-wavelet transform, Arnold transform, Hessenberg decomposition, singular-value decomposition

## Abstract

On the basis of ensuring the quality and concealment of steganographic images, this paper proposes a double-matrix decomposition image steganography scheme with multi-region coverage, to solve the problem of poor extraction ability of steganographic images under attack or interference. First of all, the cover image is transformed by multi-wavelet transform, and the hidden region covering multiple wavelet sub-bands is selected in the wavelet domain of the cover image to embed the secret information. After determining the hidden region, the hidden region is processed by Arnold transform, Hessenberg decomposition, and singular-value decomposition. Finally, the secret information is embedded into the cover image by embedding intensity factor. In order to ensure robustness, the hidden region selected in the wavelet domain is used as the input of Hessenberg matrix decomposition, and the robustness of the algorithm is further enhanced by Hessenberg matrix decomposition and singular-value decomposition. Experimental results show that the proposed method has excellent performance in concealment and quality of extracted secret images, and secret information is extracted from steganographic images attacked by various image processing attacks, which proves that the proposed method has good anti-attack ability under different attacks.

## 1. Introduction

With the development of communication technology, multimedia has become one of the most important means for people to transmit messages. At the same time, with the development of cryptography and people’s attention to security, multimedia information security has become a more concerned field. Information hiding technology plays a deep and extensive role in multimedia information security. Image is an important medium for information transmission. In order to protect the security of data, people often use image encryption technology to encrypt sensitive pictures into meaningless noise pictures. However, the encryption operation exposes the existence of sensitive pictures and is more likely to attract attacks. Compared with image encryption technology, image steganography technology hides the existence of the secret information itself and avoids the attention and suspicion of the third party. Image steganography has the advantages of protecting secret information content and hiding transmission behavior, which make the transmitted secret information more secure and confidential. The rise of image steganography makes up for the deficiency of image encryption and provides technical support for covert communication.

Steganography is defined as the science and art of hiding secret information in a way that only the intended recipient can detect the existence of the secret information [1,2]. Secret information is hidden in publicly accessed objects. These publicly accessed objects are called carrier objects. Some examples of carrier objects are audio, video, text, and images, etc., which are used in different steganography methods [3,4,5,6,7,8,9]. Among different types of carrier objects, images are considered to be excellent carriers because they have a high degree of pixel value redundancy, and the human visual system is insensitive to image details. The original image without secret information is called the cover image or the host image, and the image with secret information is called the steganographic image [1]. According to different embedding domains, existing image steganography algorithms can be divided into spatial steganography methods based on spatial domain and transform domain steganography methods based on transform domain. 

Data hiding techniques in spatial domain are mainly divided into methods based on Least Significant Bit (LSB) substitution, methods based on Pixel Value Differencing (PVD), methods based on Quantization Index Modulation (QIM), and methods based on LSB matching. In this category, LSB substitution is the most commonly used method. In this method, in order to hide the secret data in the cover image, K LSBs of the pixel value of the cover image are directly replaced with K secret message bits, so as to generate a steganographic image [10,11]. In the PVD-based method, the difference between consecutive pixels is calculated to determine the number of bits to be embedded in the cover image [12]. In LSB matching, the LSB of the cover image pixel matches the secret message bit. In the case of no matching, the pixel value of the cover image will randomly increase or decrease [13]. These techniques have very little computational overhead and are very simple. However, when more bits are embedded in each pixel, the hidden image will lose its quality and has low robustness to frequency-based attacks such as filtering and compression. In the spatial domain, many steganography schemes based on LSB are proposed to improve the quality and security of hidden images. Chang et al. [14] used the dynamic programming strategy to select the optimal replacement matrix and convert the confidential data into another value, which is then embedded in the least significant bit of the cover image. Chang et al. [15] also considered a dynamic programming strategy using modular functions to generate alternative data embedded in cover images. Bedi et al. [16] proposed an efficient spatial domain data hiding method, using the particle swarm optimization algorithm (PSO) to find the best pixel position to hide one image in another image by optimizing the similarity index. These authors extend the techniques in reference [17] and use PSO to find the best pixel location. 

In recent years, with the increasing attention and investment in covert communication technology all over the world, the means of steganography detection and analysis are becoming more and more sophisticated. The existing steganography algorithms are vulnerable to image processing attacks that damage image quality. In addition, in order to limit the channel capacity and not be suspected by third parties, the sender will actively process the steganographic image by adding noise or compression. In the image steganography technology based on the transform domain, secret information is embedded in the transform coefficients of the image. The data hiding method based on the transform domain overcomes the shortcomings of poor extraction ability and incompressibility against attacks in spatial domain technology. In this case, the host image is converted into the frequency domain by calculating the frequency coefficient of the host image. Then, the secret information is embedded by these coefficients, and the inverse transform is used to convert the image from the frequency domain to the spatial domain. Some common transform domain technologies include methods based on DFT (Discrete Fourier transforms), methods based on DCT (Discrete Cosine Transform), methods based on DWT (Discrete Wavelet Transform), and methods based on IWT (Integer Wavelet Transform) [8,18,19]. Due to the widespread use of JPEG images, most of the steganography algorithms based on the transform domain choose the DCT domain of the JPEG image as the embedding domain. The image steganography in the transform domain is developed with the embedding of the discrete cosine transform (DCT) domain. In 2016, Rabie et al. [20] proposed a DCT domain-embedding algorithm based on a global adaptive region. In [21], Change proposed a modified quantization table that embeds encrypted information in a verified transmission domain with quantized DCT coefficients. In 2016, Saidi et al. [22] discussed another steganography scheme based on DCT. The DCT coefficient is obtained by decomposition of the cover image. Chaotic mapping is used to select the embedding position, and the secret information is embedded bit by bit. The results show that the scheme has good concealment. In 2021, Dai et al. proposed a steganography algorithm based on DCT domain quantization-table modification and image scrambling [23]. 

The research on steganography algorithms that are more robust and can extract more complete secret information from the attacked steganographic image has important practical significance and application value for privacy protection and information security. With the development of JPEG2000, Discrete Wavelet Transform (DWT) has been widely used in many image processing applications. Chan et al. [24] proposed a steganography algorithm, which uses Haar discrete wavelet transform to convert spatial coverage image into frequency domain coverage image. Ghebleh et al. [25] proposed a robust chaotic algorithm based on three-dimensional chaotic Cat mapping and DWT for three-dimensional chaotic images. Ramu et al. [26] proposed a DWT-based steganography method that uses singular values to embed secret information and uses ant colony optimization to find a compromise between imperceptibility and robustness. In 2017, Miri proposed an adaptive image steganography technology based on DWT transform domain using genetic algorithm, in which the transformation of cover image is more compatible with human visual system [27]. Compressed sensing is applied to secret data, and its singular value is embedded into the low-frequency sub-band of the cover. For large-scale multimedia applications, Li et al. proposed an image steganography technology that uses cosine transform to obtain an image steganography capacity of 21.5 bpp when the signal-to-noise ratio is 38.24 dB [28]. Another high-capacity image steganography scheme is proposed by Rabie et al. [29], which uses the multi-scale Laplacian pyramid of the cover image in wavelet domain to realize data embedding, and they use the method of curve fitting adaptive region to find the appropriate hiding position in the DCT domain. Thanki et al. proposed an image steganography technique based on Finite Ridge Transform (FRT) and Discrete Wavelet Transform (DWT). Applying Arnold scrambling to secret images, this hybrid scheme proves the results in terms of imperceptibility, robustness, and computational complexity [30]. In 2020, Sharm proposed a safe, higher embedding capacity based on Discrete Wavelet Transform technology. Before embedding the correlation between the cover and the secret image, increase by multiplying some variables (1/k) by the secret image [31]. In 2021, Chudhary et al. proposed an image steganography scheme based on spectral wavelet using SVD and Arnold transform, which improved the visual quality of steganography images [32].

In order to solve the problem of poor quality of secret images extracted from steganographic images after image processing attacks, this paper proposes a dual-matrix decomposition image steganography scheme based on wavelet transform with multi-region coverage. (1) Transform the cover image to the transform domain based on the discrete wavelet transform and select the appropriate embedding area. (2) Perform Hessenberg matrix decomposition and SVD transformation on the selected transformation area to obtain singular values for embedding the secret. (3) The secret image is transformed by Arnold transform, and the secret information of the transformed image is embedded into the cover image. Multi-level discrete wavelet transform is used to protect the hidden secret information and reduce the loss of secret information when the steganographic image is attacked by image processing attack. In this paper, multi-level discrete wavelet transform and double-matrix decomposition are used to ensure the robustness of the steganographic scheme, and more complete secret information can be extracted from the attacked steganographic image. The hidden area with multi-region coverage ensures the concealment of the scheme so that it is not suspected by the third party and protects the security of secret information. A variety of image-processing attacks are performed on the steganographic image to check the robustness of the proposed scheme. 

The arrangement of this paper is as follows: Section 2 introduces the theoretical methods applied in this paper. Section 3 introduces the image steganography scheme, including embedding method and extraction method. Section 4 presents the results demonstration, performance analysis, and robustness test of the proposed scheme. Section 5 concludes. 

## 2. Materials and Methods

### 2.1. Image Wavelet Transform

The translation factor and scale factor in continuous wavelet transform are real numbers of continuous transformation. Therefore, it is necessary to discretize the continuous wavelet transform, retain more information as much as possible through less calculation, and obtain wavelet coefficients with lower redundancy, namely discrete wavelet transform.

In image processing, most of the signals are two-dimensional, even multi-dimensional. Therefore, wavelet analysis theory is more often used in the processing of two-dimensional or multi-dimensional signals. For two-dimensional discrete wavelet transform, it is defined as follows:(1)$$(W_{\varphi }f)(a,b_{1},b_{2})=\int_{-\infty }^{+\infty }\int_{-\infty }^{+\infty }f(t_{1},t_{2})\frac{1}{a}\varphi_{a,b}(\frac{\left(t_{1},t_{2})-(b_{1},b_{2}\right)}{a})dt_{1}dt_{2}$$

The inverse transformation is:(2)$$f(t_{1},t_{2})=\frac{1}{C_{\varphi }}\int_{-\infty }^{+\infty }\int_{-\infty }^{+\infty }\int_{-\infty }^{+\infty }\frac{1}{a}(W_{\varphi }f)(a,b_{1},b_{2})\varphi_{a,b}(t_{1},t_{2})dadb_{1}db_{2}$$

The application of wavelet transform in digital image is to decompose the image in two dimensions, and finally place sub-images in different spaces and different frequencies. The lower the resolution of the sub-image is, the more information it can contain.

After the image is decomposed by wavelet transform, the absolute value of the low-frequency sub-band coefficients is largest, and most of the energy of the image is concentrated in this part. The low-frequency sub-band can be further decomposed, and the transformed low-frequency sub-band is similar to the cover image. Compared with other transforms, the image still has the characteristics of the spatial domain and frequency domain of the original image after wavelet transform [33]. The low-frequency coefficients correspond to the main features of the image, and the high-frequency part contains the information of the edge and texture of the image, as shown in Figure 1.

The decomposition of the image by wavelet transform is to characterize the image in different frequency bands and spatial directions [34]. The sensitivity of human eyes to different frequency bands is different. According to the different sensitivity of human vision to different frequency bands, we choose the most suitable position for image steganography to embed secret information in the image, so that the steganographic image has better concealment. At the same time, the image steganography technology in the wavelet domain can effectively resist the attacks of clipping, compression, and filtering.

Therefore, using wavelet transform to process the image to embed secret information can make steganography technology able to resist various attacks on the premise of guaranteeing the original quality of the image, and at the same time ensure the integrity of extracted secret information.

### 2.2. Arnold Transformation

Arnold transform was proposed by Vladimir Igorevich Arnold, a Russian mathematician. Arnold transform can replace the position of each pixel in the image to achieve the purpose of encryption, so it is mostly used in multimedia such as images to improve safety [35].

The specific expression of Arnold transformation is:(3)$$\left[\begin{array}{c} x_{n+1}\\ y_{n+1} \end{array}\right]=\left[\begin{array}{cc} 1 & b\\ a & ab+1 \end{array}\right]\left[\begin{array}{c} x_{n}\\ y_{n} \end{array}\right]\mathrm{mod}(N)$$
where *x*, *y* represent the position of the pixel in the grayscale image before transformation; *x**_n+_*_1_, *y_n+_*_1_ represent the pixel position after transformation; *a*, *b* are parameters and both are positive integers; *n* represents the number of current transformations; *N* is the length or width of the image, and mod is Modular operation.

In particular, when *a* = *b* = 1 and *N* = 1, the Arnold transformation can be written as:(4)$$\left[\begin{array}{c} x_{n+1}\\ y_{n+1} \end{array}\right]=\left[\begin{array}{cc} 1 & 1\\ 1 & 2 \end{array}\right]\left[\begin{array}{c} x_{n}\\ y_{n} \end{array}\right]\mathrm{mod}1=C\left[\begin{array}{c} x_{n}\\ y_{n} \end{array}\right]\mathrm{mod}1$$

The two reasons that Arnold mapping can produce chaos are stretching and folding. Stretching refers to multiplying by matrix *C* to increase the value of *x*, *y*, and folding refers to taking the modulus to fold *x*, *y* back into the unit matrix. Due to xwthe two properties of stretching and folding, the two adjacent points are no longer adjacent after many iterations of transformation, which can be used to fully disrupt the position of the adjacent pixels of an image. The relevant information of the original image cannot be obtained from the image so that the purpose of protecting the image information can be achieved. It should also be noted that the Arnold transform is periodic, and it will return to the original state after many Arnold transformations. The number of iterations to return to the original state will be different with different image sizes, so we pay attention to the number of iterations when selecting images of different sizes.

### 2.3. Hessenberg Matrix Decomposition

Hessenberg matrix decomposition is a matrix decomposition method that is often used to decompose square matrices. If the elements of the matrix *H* = (*h_ij_*)*_n×n_*∈*R^n×n^* satisfy *h_ij_* = 0(*j* > *i* + 1), then *H* is called the upper Hessenberg matrix, and its specific form is:(5)$$H=\left[\begin{array}{cccc} h_{11} & h_{12} & \cdots & h_{1n}\\ h_{12} & h_{22} & \vdots & \ddots \\ & \ddots & \ddots & h_{n-1,n}\\ 0 & \cdots & h_{n,n-1} & h_{nn} \end{array}\right]$$

Suppose *X* = (*a_ij_*)∈*R^M×N^*, then there are orthogonal matrices *Q*_1_, *Q*_2_, …, *Q_n_*_−2_, so that *X* is transformed into an upper Hessenberg matrix through orthogonal similar transformation, that is *Q_n−_*_2_, …, *Q*_2,_
*Q*_1_*X Q*_1_, *Q*_2_, …, *Q_n_*_−2_ = *H*. From the above, we can acquire the calculation formula of Hessenberg matrix decomposition as follows:(6)$$PHP^{T}=HD(X)$$
where *P* is an orthogonal matrix, *H* is a Hessenberg matrix, and for any *i* > *j* + 1, the element *h_i,j_* of *H* is 0.

Hessenberg is usually calculated by the Houthor matrix, and the formula is as follows:(7)$$Q=(I_{n}-2\mu \mu^{T})/\mu^{T}\mu$$
where μ is a nonzero vector and *I_n_* is the identity matrix of *n×n*.

Combining the above two formulas, the Hesseberg decomposition can be evolved into:(8)$$\begin{array}{l} P={\left(Q_{1}Q_{2}\ldots Q_{n-2}\right)}^{T}X(Q_{1}Q_{2}\ldots Q_{n-2})\\ \Rightarrow H=P^{T}XP\\ \Rightarrow X=PHP^{T} \end{array}$$

Suppose *A* is an image block of size 4 × 4 as the formula (9), and after Hesseberg decomposition, that is [*P*,6*H*] = *hess*(*A*), we can acquire a unit matrix *P* and an upper triangular Hessenberg matrix *H* such as (10) and (11). In the *H* matrix, the value of *H*(2,2) is the largest, which represents most of the energy of the image block. Experimental studies have found that the *H*(2,2) element has a certain numerical stability to external disturbances. From this perspective, the Hessenberg matrix decomposition of the image can achieve a more stable part of the image [36]. In image steganography technology, the cover image is decomposed by Hessenberg matrix, and the secret information is embedded in the matrix obtained after decomposition, which can effectively improve the anti-attack performance.
(9)$$A=\left[\begin{array}{cccc} 124.000 & 132.000 & 135.000 & 142.000\\ 112.000 & 142.000 & 154.000 & 161.000\\ 58.000 & 69.000 & 82.000 & 100.000\\ 102.000 & 82.000 & 70.000 & 66.000 \end{array}\right]$$
(10)$$P=\left[\begin{array}{cccc} 1 & 0 & 0 & 0\\ 0 & -0.6905 & -0.5183 & -0.5046\\ 0 & -0.3576 & -0.3168 & 0.8610\\ 0 & -0.6228 & 0.7749 & 0.0645 \end{array}\right]$$
(11)$$H=\left[\begin{array}{cccc} 124.000 & -228.7039 & -7.2315 & 58.7812\\ -162.2097 & 303.0602 & 9.3364 & -78.8986\\ 0 & 87.0072 & -14.9255 & -34.3097\\ 0 & 0 & 11.6062 & 1.8653 \end{array}\right]$$

### 2.4. Singular-Value Decomposition

Singular-value decomposition is a method of matrix analysis in which the matrix is processed based on the eigenvectors of the matrix. The singular vector of the matrix can be obtained through singular-value decomposition, maintaining the correlation of the rows or columns of the original matrix.

The singular-value decomposition of a matrix means that for a non-zero real number matrix A of any size *M × N*, it can be expressed as the product of two-unit orthogonal matrices and a diagonal matrix [37]. The singular-value decomposition can be written in the following form:(12)$$A=USV^{T}$$
where *S* is a singular-value matrix, *S* = diag$\left(\sigma_{1},\sigma_{2},\cdots ,\sigma_{r},0,\cdots ,0\right)$, the diagonal elements of *S* matrix are r singular values of matrix *A*, *U* and *V* are two orthogonal matrices, and each column element of *U* and *V* represents the left singular vector and the right singular vector, respectively—that is, $Av_{i}=\sigma_{i}u_{i}$, $Au_{i}=\sigma_{i}v_{i}$.

Singular-value reconstruction is to inversely transform the singular-value matrix *S* and two diagonal matrices to form a new matrix. The secret information can be hidden by modifying the singular value in the singular-value matrix to form a new singular-value matrix to embed the secret information.

The singular value of the singular-value matrix represents the characteristic information of the image. The diagonal elements of the singular-value matrix after matrix transformation are arranged from large to small, and the value of the first diagonal element is larger than the other diagonal elements. Ignoring other singular values will hardly affect the reconstructed image. In addition, when the image is slightly disturbed, the diagonal elements of the singular-value matrix obtained through singular-value decomposition will not change significantly, and the singular-value decomposition of the matrix has the characteristics of rotation invariance. From this aspect, embedding the secret image after the singular-value decomposition of the image can improve the anti-attack performance of the algorithm.

### 2.5. Logistic Chaotic System

Chaos is an unstable state under certain conditions. Chaotic system has become one of the most popular tools in image encryption technology because of its uncertain chaotic behavior, unpredictable trajectory, universality, ergodicity, and many other excellent characteristics. 

Logistic chaotic system is a classical chaotic system. Logistic chaotic map is very simple in mathematical form, but it has extremely complex dynamic behavior [38]. Therefore, Logistic chaotic system is widely used in data security and secure communication. The mathematical expression of the logistic equation is:(13)$$x_{n+1}=\mu x_{n}(1-x_{n})$$

𝜇 is the system parameter of the logistic equation, and the initial value of the system is set to 𝑥_0_(0 < 𝑥_0_ < 1). When 3.5699 < 𝜇 < 4, the system goes into chaos.

## 3. A Steganography Scheme of Dual-Matrix Decomposition Image with Multi-Region Coverage

This section mainly introduces the proposed dual-matrix decomposition image steganography scheme based on wavelet transform with multi-region coverage. It is mainly introduced from four aspects: (1) Select the embedding area of the cover picture. (2) The method and process of embedding the secret image. (3) The extraction process of the secret image. (4) The logistic chaotic encryption algorithm encrypts the key.

### 3.1. Hidden Region Selection in Wavelet Domain

There are many algorithms for embedding secret images into cover images. From the perspective of embedding, they are mainly divided into two different methods: spatial domain and transform domain. When these methods are implemented, they all need to embed the secret image through the coefficients of some transformations, but in the end, they all achieve the ultimate goal of embedding secret information by changing the grayscale values of some pixels. The image steganography scheme proposed in this paper is to perform multi-discrete wavelet transform on the cover image and embed the secret image in the selected hidden area in the wavelet domain of the cover image.

In this paper, the cover image is transformed by two-stage discrete wavelet transform. As shown in Figure 2, the first-order low-frequency region contains most of the image energy corresponding to the main features of the image and the other parts contain the image edge and texture information, and the second-order component LL2 also contains most of the energy of the first-order component. In this paper, the region shown in Figure 3 is selected as the embedded region, which is the middle part of the two-level discrete wavelet transform image, including the sub-band HH2 and some edge and texture information with a small contribution rate. Modifying the coefficient value of the selected hidden region has little impact on the overall image. Embedding the secret image in the selected region ensures the robustness and concealment of the steganography scheme.

### 3.2. Steganographic Embedding Algorithm for Secret Image

The input of the steganography algorithm is the cover image *C*, the secret image *S*, and embedding factor α. The output image is the steganographic image *C** with secret information. The size of the cover image is *M* × *M*, the size of the secret image is *M*/2 × *M*/2, and the size of the output image with secret information is *M* × *M*. In this paper, Hessenberg matrix decomposition and singular-value decomposition are used to obtain the steganographic image in the embedding process. In addition, this paper selects an appropriate embedding strength factor α to ensure that the steganographic algorithm has better performance in terms of invisibility and robustness. The steps of secret image embedding are shown in Figure 4. The specific embedding algorithm is Algorithm 1, as follows:
**Algorithm 1:** The embedding algorithmInput: Cover image ***C***, Secret image ***S***, embedding factor ***α***.Output: Steganographic image ***C****, Encrypted key matrix ***U_w_*_1*′*_**, ***V_w_*_1_*‘***.1: [*LL2*, *HL2*, *LH2*, *HH2*, *HL*, *LH*, *HH*] = DWT2level(*C*)2: *RS* = [*HH*2, *HL*(1:128,129:256);*LH*(129:256,1:128), *HH*(1:128,1:128)]3: *RS’* = Arnold (*RS*)4: *P H P^T^* = Hessenberg (*RS*’)5: *HU_w_* ⋅ *HS_w_*
⋅
*HV_w_* = SVD (*H*)6: *S’* = Arnold (*S*)7: *U_w_*_1_
*S_w_*_1_
*V_w_*_1_*^T^* = SVD (*S’*)8: HSw1*=HSw+α*S_w_*_1_
9: *H** = *HU_w_ HS_w_*_1_** HV_w_*10: *RSS’* = *P H*P^T^*11: *RSS* = Rearnold(*RSS’*)//The hidden area after embedding secret information is shown in the Figure 5. In Figure 5, HL*, LH*, HH* and HH2* are HL, LH, HH, and HH2 embedded with secret information respectively.12: *HH2** = *RSS* (1:128,1:128)13: *HL** = *HL*, *LH** = *LH*, *HH** = *HH*14: *HL**(1:128,129:256) = *RSS* (129:256,1:128)15: *LH**(129:256,1:128) = *RSS* (1:128,129:256)16: *HH**(1:128,1:128) = *RSS* (129:256,129:256)17: *C** = IDWT2level [*LL2*, *HL2*, *LH2*, *HH2**, *HL**, *LH**, *HH**]18: (*U_w_*_1*′*_, *V_w_*_1_*‘*) = ChaoticEncrypt ( *U_w_*_1_, *V_w_*)

Among them, *U_w_*_1_ and *V_w_*_1_ are the keys when extracting secret images. To ensure the security of the keys, the encryption algorithm of logistics chaos system is used to encrypt them to obtain *U_w_*_1*′*_ and *V_w_*_1_*‘*.

In the embedding algorithm, the secret information is multiplied by the embedding strength factor α and then added to the singular value obtained by the two-level matrix decomposition of the selected hidden area to obtain the singular value of the steganographic image, which achieves the purpose of embedding secret information. The larger the value of α, the greater the difference between the singular value after embedding the secret information and the original singular value and the greater the difference between the steganographic image and the original cover image. However, the larger the value of α, the stronger the ability to extract the secret information from the attacked steganographic image when the steganographic image is attacked. The smaller the value of α, the smaller the difference between the singular value after embedding the secret information and the original singular value, and the smaller the difference between the steganographic image and the original cover image. However, the small value of α has poor ability to extract the secret information from the attacked steganographic image when the steganographic image is attacked, so it cannot recover the high-quality secret image. In other words, the larger the value of α, the stronger the anti-attack performance and the worse the concealment of secret information; the smaller the value of α, the worse the anti-attack ability of steganographic image and the better the concealment of secret information. Only selecting the appropriate embedding factor can ensure the ability of extracting secret information from the attacked image and the high concealment of secret information.

The embedding algorithm uses singular-value decomposition and Hessenberg matrix decomposition. The elements in the matrix obtained by singular-value decomposition and Hessenberg matrix decomposition have a certain stability when subjected to external disturbance, which improves the robustness of the steganography scheme. The appropriate embedding strength factor α and the hidden area with multi-region coverage ensure the concealment of secret information. Also, the embedding algorithm has low complexity and is easy to implement and apply.

### 3.3. Steganographic Extraction Algorithm of Secret Image

In the secret image extraction algorithm, the input is the cover image embedded with secret information—steganographic image *C**, embedding factor α and the key *U_w_*_1_, *V_w_*_1_, and the output is the extracted secret image—extracted image. The overall process of the extraction algorithm is shown in Figure 6, and the specific extracting algorithm is Algorithm 2, as follows:
**Algorithm 2:** The extracting algorithmInput: Steganographic image ***C****, Encrypted key matrix ***U_w_*_1*′*,_*V_w_*_1_*′***.Output: The extracted Secret image ***C’***.1: [*LL_w_*_2_, *HL_w_*_2_, *LH_w_*_2_, *HH_w_*_2_, *HL_w_*, *LH_w_*, *HH_w_*] = DWT2level (*C**)2: *RS_w_* = [*HH_w_*_2_, *HL_w_*(1:128,129:256); *LH_w_*(129:256,1:128), *HH_w_*(1:128,1:128)]2: *RS_w_** = Arnold (*RS_w_*)3: (U_w1,_ V_w1_) = ChaoticDecrypt (*U_w_*_1*′*_, *V_w_*_1*′*_)4: *P_w_H_w_P_w_*^T^ = Hessenberg (*RS_w_**)5: *HU_w_*HSb_w_*HV_w_**^T^ = SVD (*H_w_*)6: *S*_*w*1_*** = (*HSb*_*w*1_*-*HS_w_*)/*α*7: *C*1*′* = *U_w_*_1_
*S_w_*_1_** V_w_*_1_*^T^*8: *C’* = Rearnold (*C*1′)

### 3.4. Key Encryption Algorithm of Logistic Chaotic Map

The key is very important for image extraction. Only when the security of the key is ensured can the secret image be successfully extracted. Chaotic system is a cryptographic system with uncertain, nonlinear, and random behavior. It has many characteristics, such as ergodicity, sensitivity to the conditions of initial value occurrence, strong unpredictability, and randomness. Using chaotic system to encrypt the key can ensure the randomness and security of the encrypted key.

The steps of the logistic chaotic encryption algorithm for key encryption are as follows:

Step 1. Determine the parameters and initial values of the logistic chaotic mapping and substitute them into the equation of the Logistic mapping to iterate *M × N* times (*M × N* is the size of the key matrix, *M* is the length and *N* is the width) and generate the one-dimensional floating point chaotic sequence *CL* with the length of *M × N* and the value between (0,1).

Step 2. Each element in the sequence *CL* is transformed by multiplying 255 to (0, 255), then the integer sequence between (0, 255) is obtained, and the new integer sequence is transformed into an *M × N* two-dimensional matrix with the same size as the key matrix.

Step 3. XOR the corresponding bits of the two-dimensional matrix obtained through the chaotic system and the key matrix to obtain the encrypted key.

## 4. Simulation Results and Analysis

### 4.1. Evaluation Indicators

The purpose of image steganography technology is to conceal the existence of secret image information so as not to suffer the attention and suspicion of a third party, to achieve the purpose of covert transmission or storage and ensure the security of secret information. Therefore, the concealment of the steganographic image embedded with the secret image information is an important index to evaluate the performance of the steganography algorithm. The concealment degree of the steganography image is measured by the two parameter values of peak signal-to-noise ratio and structural similarity.

Peak signal-to-noise ratio (PSNR) is an objective evaluation index to compare two images. This evaluation method is an evaluation index produced by image error sensitivity. The formula of PSNR is:(14)$$\mathrm{PSNR}(C,C^{*})=10\mathrm{lg}\frac{C_{\max}^{2}}{\mathrm{MSE}}$$
(15)$$\mathrm{MSE}=\frac{1}{M^{2}}\sum_{i=1}^{M}\sum_{j=1}^{M}{\left(C_{i,j}-C_{i,j}^{*}\right)}^{2}$$
where MSE is the mean square error between the cover image and the steganographic image embedded with secret information, and *C*_max_ is the maximum pixel value in the cover image.

When the peak signal-to-noise ratio is greater than 37 dB, it is invisible to the human visual system, and when the structural similarity is greater than 0.93, it shows that there is little difference between the cover image and the steganography image embedded with secret information [39].

Structural similarity (SSIM) is an image quality evaluation index based on structural similarity theory. The image is highly structured and has a strong correlation between pixels. When measuring the distance between two images, humans pay more attention to structural similarity than calculating the difference per pixel [40]. The formula of SSIM is:(16)$$SSIM=\frac{\left(\mu_{C}\mu_{C*}+d_{1})(\sigma_{CC*}+d_{2}\right)}{\left(\mu_{C}^{2}+\mu_{C*}^{2}+d_{1})(\sigma_{C}^{2}+\sigma_{C*}^{2}+d_{2}\right)}$$
where $\mu_{C}$ and $\mu_{C*}$ are the average of *C* and *C**, $\sigma_{C}^{2}$ and $\sigma_{C*}^{2}$ are the variance of *C* and *C**, $\sigma_{CC*}$ is the covariance of *C* and *C**, and *d*_1_ and *d*_2_ are two variables which are uesd to stabilize the division with a weak denominator.

When the peak signal-to-noise ratio is greater than 37 dB, it is invisible to the human visual system, and when the structural similarity is greater than 0.93, it shows that there is little difference between the cover image and the steganography image embedded with secret information [41].

For the purpose of not being suspected by a third party, the sender actively scrambles the steganography image or the third party attacks the steganography image in the process of transmission, which will cause the loss of steganography image quality. The ability to extract secret images from the attacked steganography images is also the focus of the analysis. The normalization coefficient (NC) is often used to evaluate the integrity between the extracted image and the original image. The normalization coefficient reflects the integrity of the secret information extracted from the secret image, and the integrity of the secret information is expressed by the corresponding numerical value [42]. The expression of NC is as follows:(17)$$NC=\frac{\sum_{i=1}^{M}\sum_{j=1}^{N}S(i,j)S{\left(i,j\right)}^{*}}{\sqrt{\sum_{i=1}^{M}\sum_{j=1}^{N}S{\left(i,j\right)}^{2}}\sqrt{\sum_{i=1}^{M}\sum_{j=1}^{N}S(i,j){*}^{2}}}$$

The normalization coefficient (NC) is used to evaluate the integrity of the secret image extracted from the attacked image to verify the robustness of the proposed scheme.

### 4.2. Parameter Setting

The experiment in this article is carried out on a computer equipped with Intel quad-core 3.7 Ghz and 16.0 GB RAM, using MATLAB R2020b software for simulation. In the process of experiment, subjective vision and objective quantitative analysis are used to analyze the invisibility and robustness of the algorithm, and attacks with different parameters are used to further evaluate the robustness of the proposed algorithm.

The experiment uses a 512 × 512 Lena image as the cover image as shown in Figure 7a, and this paper uses a 256 × 256 parrot image as the secret image as shown in Figure 7b for steganography. First, we find the most suitable embedding strength factor α through PSNR, SSIM, and NC curves under different attacks. The choice of embedding intensity factor α is very important to the performance of the algorithm. Figure 8a–c shows the change curves of PSNR value, SSIM value, and NC value under different attacks of steganographic images with the change of embedding strength factor. As can be seen from Figure 8, with the gradual increase of the embedding intensity factor, the higher the normalized correlation of the extracted secret information, the better the integrity of the extracted secret image. However, with the increase of the embedding intensity factor, the PSNR value and SSIM value of the steganographic image under various attacks gradually decrease, and the appropriate embedding intensity factor can ensure the high concealment of the steganographic image and extract the secret image completely at the same time. The performance of the algorithm can achieve a good balance between concealment and the ability to extract secret information from the attacked image. It can be seen from Figure 8 that the concealment of the steganography image reaches an optimal state when the value of the embedding factor is in the range of 0:0.02. Meanwhile, the secret image extracted from various attacks in the range of 0:0.02 can also ensure better integrity. According to the above analysis, the value of embedding intensity factor is selected as 0.01 in the proposed scheme to ensure that the secret information can be completely extracted and the concealment of steganography image can be guaranteed. We use different mother wavelets to experiment the steganography scheme proposed in this paper. We chose Lena image as the cover image and parrots as the secret image. When the mother wavelets are Haar and db, PSNR is 49.5106 db, SSIM is 0.99389, and NC is 0.99955 in the experimental results. When the mother wavelets are sym, bior, and corif, etc., the size of the cover image changes after multi-wavelet transform, which is not suitable for the steganography scheme in this paper. The type of mother wavelet determined in this paper is Haar.

When testing the robustness of the algorithm, various image attacks are added to steganographic image, including noise, clipping, compression, rescaling, histogram equalization, sharpening, and rotation attacks. Table 1 lists the attack types and parameters. The various parameters of the attack are as follows: the noise attack is a noise addition with a density or variance of 0.001, the cropping attack is a clipping with a clipping ratio of 2%, a JPEG compression attack with a quality factor of 50 and a JEPG2000 compression attack with a compression ratio of 12, rescaling coefficients of scaling attack are 0.25 and 4, motion blur parameters are 4 and 7, sharpening attack with threshold value of 0.8, and rotation angle of rotation attack is 2.

### 4.3. Concealment Analysis

In order to ensure the security of the secret image, the human visual system cannot see the trace of the secret image from the steganography image embedded with the secret information. In this paper, the secret image is embedded into the cover image to obtain the steganographic image, and the secret image is extracted from the steganographic image without any attack. The simulation results are shown in Figure 9 and Figure 9a is the steganography image embedded with the secret image information. Figure 9b is the extracted secret image and shows the peak signal-to-noise ratio and structural similarity between the steganographic image and the original cover image. When the peak signal-to-noise ratio is greater than 37 dB, the secret information embedded in the cover image is invisible to the human visual system, and when the structural similarity is greater than 0.93, it indicates that the steganographic image is not much different from cover image.

As can be seen from Figure 9, the PSNR of the Lena steganographic image embedded with the secret image of the parrot is 49.5106 dB and the SSIM value is 0.99849. From the perspective of human vision, there is no difference between the steganographic image and the original cover image, and the complete secret information can be observed from the extracted secret image. Figure 10 shows a histogram matching of the Lena image and steganographic image, which also indicates better concealment.

In addition to the Lena image, Baboon, Airplane, Peppers, Man, Tiffany, and Barbara images are selected as cover images to test the concealment of the algorithm. Each cover image is converted into a grayscale image with a size of 512 × 512, and a 256 × 256 parrot image is used as a secret image. Each cover image is converted into a 512 × 512 grayscale image, a 256 × 256 parrot image is used as the secret image, and an embedding factor α of 0.01 is selected. Figure 11 shows steganographic images embedded with secret information, and the observed steganographic image in Figure 11 has excellent visual quality. Table 2 shows the objective evaluation values using various image quality indicators.

The results show that the steganographic image obtained by the proposed steganographic method has good concealment, which meets the requirement of the human visual system that the secret image information on the cover image cannot be seen from both subjective and objective aspects. The secret image extracted from the steganographic image also has a high NC value, which ensures the integrity of the extracted secret information. The proposed algorithm selects the embedded hidden region in wavelet domain to enhance the concealment of the algorithm and compares it with the traditional singular-value embedding algorithm in the DWT wavelet domain, which chooses a low-frequency sub-band as the hidden region. The results are shown in Table 3.

According to Table 3, compared with the traditional DWT-SVD scheme [43], the PSNR value of the scheme proposed in this paper has been significantly improved and the scheme still has excellent structural similarity. Through the horizontal comparison with the concealment of other steganography schemes, Table 4 shows the comparison of PSNR, SSIM, and NC indexes when the cover images are Lena images. By comparing with other existing algorithms, it is proven that the proposed algorithm performs well in each performance index, the steganography image will not attract the attention of eavesdroppers, and the quality of the steganographic image is excellent, so the third party cannot suspect the existence of the hidden information.

### 4.4. Robustness Analysis

When the steganographic image is transmitted through the network, the third party may try to retrieve the secret information or destroy it. It is very meaningful to design a robust steganographic algorithm against attack. After meeting the objective requirements in concealment, it is necessary to further evaluate the robustness of the algorithm. A detailed anti-attack test is carried out on the steganography algorithm of the multi-level matrix decomposition proposed in this paper, and a variety of attack methods are used to attack the steganography image. The proposed extraction algorithm is used to extract the secret image from the attacked steganography image, and the quality of the extracted secret image is verified. Several attacks mentioned in Table 1 are carried out on the steganography image embedded with secret information. Table 5 shows that when baboon, pepper, and Lena are used as cover images, the NC values of the secret image are extracted under different attacks and the NC values are compared with the steganography algorithm using single-level singular-value decomposition. The NC value of the secret image extracted by the proposed algorithm is generally greater than the NC value of the secret image extracted by the single-stage singular-value decomposition scheme. According to the analysis of Table 5, the secret image extracted by this algorithm has stronger integrity, and the secret image extracted by the two-stage matrix decomposition proposed in this paper has stronger integrity than the secret image extracted by the single-stage singular-value matrix decomposition, which proves that the proposed scheme has better robustness.

Figure 12 shows the secret image extracted from the attacked steganography image when the cover image is a Lena. The extracted secret image is lost in information, but the extracted secret image under most attacks is still acceptable in human vision, and a large amount of information can be obtained from the extracted secret image. Except for the motion blur attack and histogram attack, the normalized correlation values of all extracted secret images are above 0.93. For noise attacks, the normalized correlations of Gaussian noise, salt and pepper noise, and speckle noise are 0.97996, 0.98663, and 0.96599, respectively; for compression attacks, the normalized correlations of JPEG compression and JPEG2000 compression are 0.94866 and 0.9815, respectively; for sharpening qttack, the normalized correlation is 0.96797; for histogram equalization, the normalized correlation is 0.88757; for the motion blur attack, the normalized correlation is 0.561.

In the above tests, the parameters of the attacks are fixed to attack the steganographic image, but considering the actual scene, the parameters of the attacks are dynamically variable, so the experimental test is carried out under the dynamic attack parameters. When the cover image is Lena and the secret image is parrot, the test results are as follows:

In Figure 13, the performance of anti-JPEG attack under different compression strength QF was tested. QF varies from 10 to 90, with a step size of 10. The smaller the compression strength value, the more compressed the image will be. When the compression strength is from 90 to 10, the normalized similarity of the extracted secret image shows a downward trend as a whole, and the normalized coefficient can still reach 0.9682 even when the compression strength is 10. In Figure 13, the normalized similarity of secret images extracted under JPEG2000 compression attacks with different compression rates was tested. The normalization coefficient showed a downward trend when the compression ratio was changed from 4 to 36, and the measured normalized similarity was greater than 0.96. When the maximum compression rate was 36, the normalized coefficient was 0.963731. At the same time, Gaussian noise with different variances and sharpening attack with different thresholds were also tested, as shown in Figure 13. When the variance of Gaussian noise is 0.001, the normalization coefficient can be as high as 0.97996. When the variance is 0.009 at most, the normalization coefficient value is still 0.82752. When the sharpening attack threshold is 0.1, the normalized correlation between secret image and extracted secret image is 0.990497, and when the sharpening attack threshold is 0.9 at most, the normalized correlation is 0.832227. It can be seen from the above results that the image steganography method proposed in this paper shows excellent robustness against Gaussian noise attacks, sharpening attacks, and JPEG and JPEG2000 compression attacks with variable parameters.

## 5. Conclusions

In this paper, a double-matrix decomposition image steganography scheme with multi-region coverage is proposed. This scheme solves the problem of poor quality secret-image extraction when the steganographic image is attacked by image processing or the sender takes the initiative to attack the steganographic image.

Firstly, the stability of the matrix obtained by Hessenberg matrix decomposition and singular-value decomposition under disturbance is introduced. The steganography scheme of this paper is proposed by using the characteristics of these two kinds of matrix decomposition. The second-order wavelet transform is performed on the cover image, and the hidden regions covering multiple sub-bands are selected in the wavelet domain of the cover image. As the matrix after Hessenberg matrix decomposition has a certain stability to the numerical disturbance, in order to enhance the robustness of the scheme, the hidden area selected in the wavelet domain is Arnold transformed and then subjected to Hessenberg matrix decomposition to obtain the matrix *H*; the matrix *H* is decomposed by singular-value decomposition, and the obtained singular-value matrix can remain stable when disturbed. The secret information is embedded into the singular-value matrix of the cover image through the embedding factor, and the steganographic image is obtained through two-level inverse matrix decomposition and inverse Arnold transformation. The PSNR value of the steganographic image is as high as 49.5106dB and the SSIM is 0.99389. Embedding the secret information into the cover image has little effect on the quality of the cover image. It has been proved that the scheme proposed in this paper has a strong invisibility. The NC value of the secret image extracted from the steganographic image is more than 0.9. Most of the information in the secret image can be extracted even if the attack intensity is strong under the dynamic attack coefficient. The robustness of the proposed steganography algorithm is proved.

The steganography scheme proposed in this paper is also suitable for when the cover image is RGB image and the secret image is a grayscale image. The secret image can be embedded into any component image of the cover image, and the grayscale image can be hidden in the color image, which also reflects the excellent concealment and robustness of the steganography scheme. When the secret image and the cover image are both color RGB images, embedding the three component images of the secret image into the three component images of the cover image will degrade the quality of each component image of the cover image. The final steganographic image is quite different from the original image and the existence of the secret image cannot be hidden, so it is easy to be suspected by the third party. This problem is also an important direction for color image steganography in the future.

## Figures and Tables

**Figure 1 entropy-24-00246-f001:**
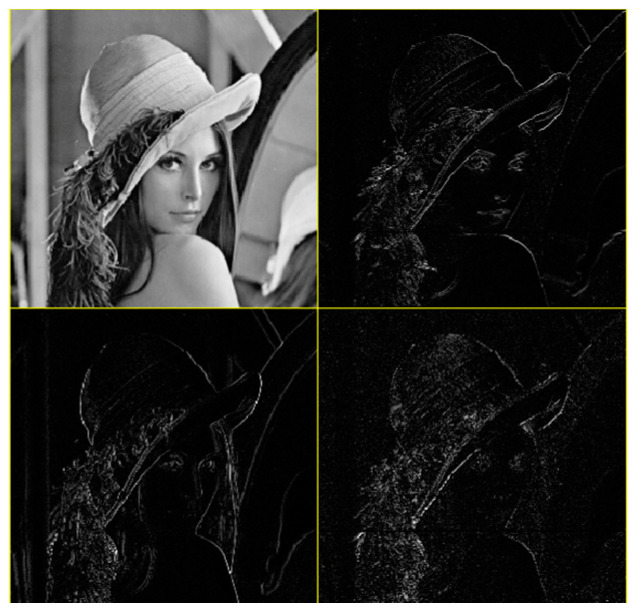
Wavelet transform Lena.

**Figure 2 entropy-24-00246-f002:**
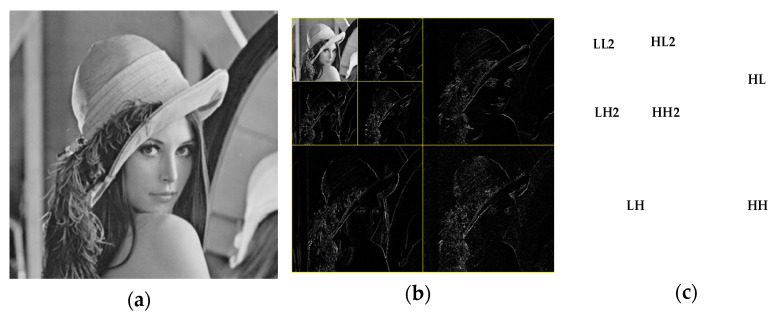
Second-order discrete wavelet transform. (**a**) Original image of Lena; (**b**) Multi-wavelet transform Lena; (**c**) Schematic diagram of component position.

**Figure 3 entropy-24-00246-f003:**
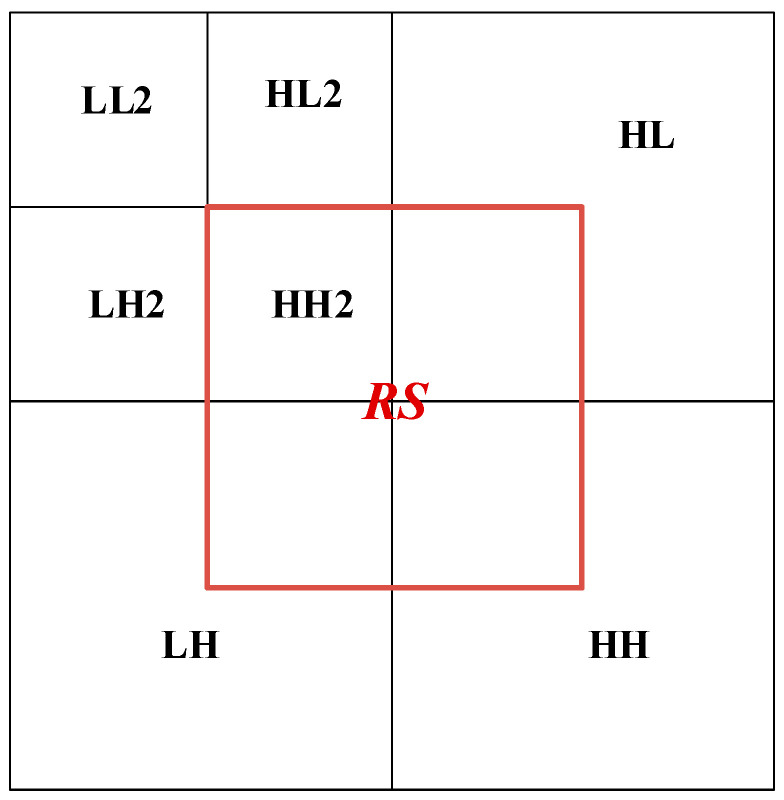
Selected hidden area.

**Figure 4 entropy-24-00246-f004:**
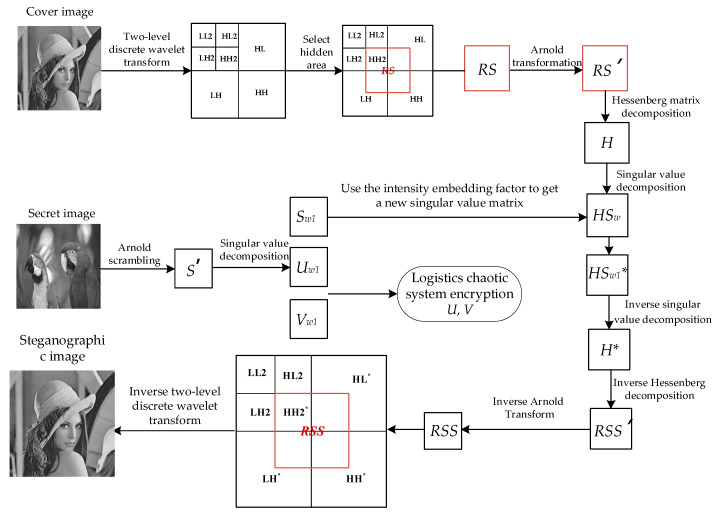
Flowchart of secret image embedding.

**Figure 5 entropy-24-00246-f005:**
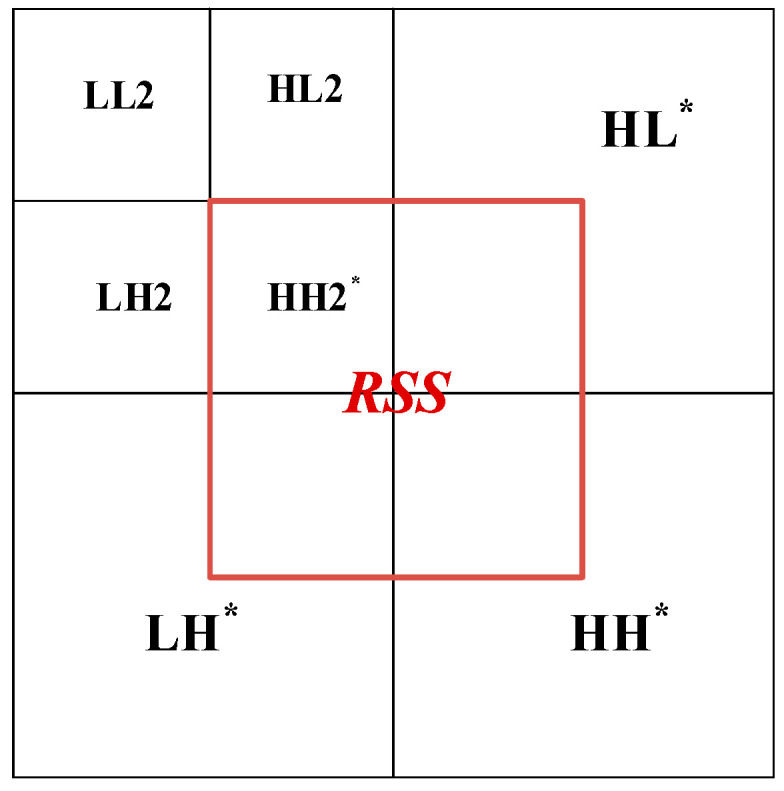
The hidden area after embedding information.

**Figure 6 entropy-24-00246-f006:**
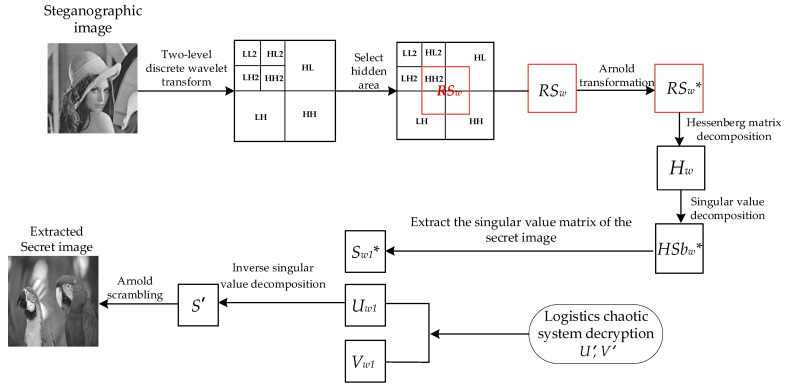
Flow chart of the secret image extraction process.

**Figure 7 entropy-24-00246-f007:**
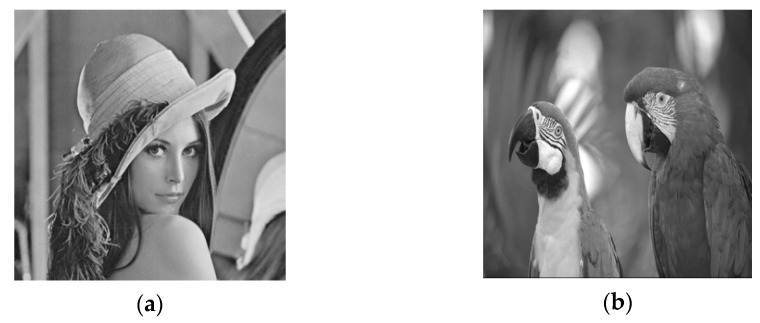
(**a**) cover image (**b**) secret image.

**Figure 8 entropy-24-00246-f008:**
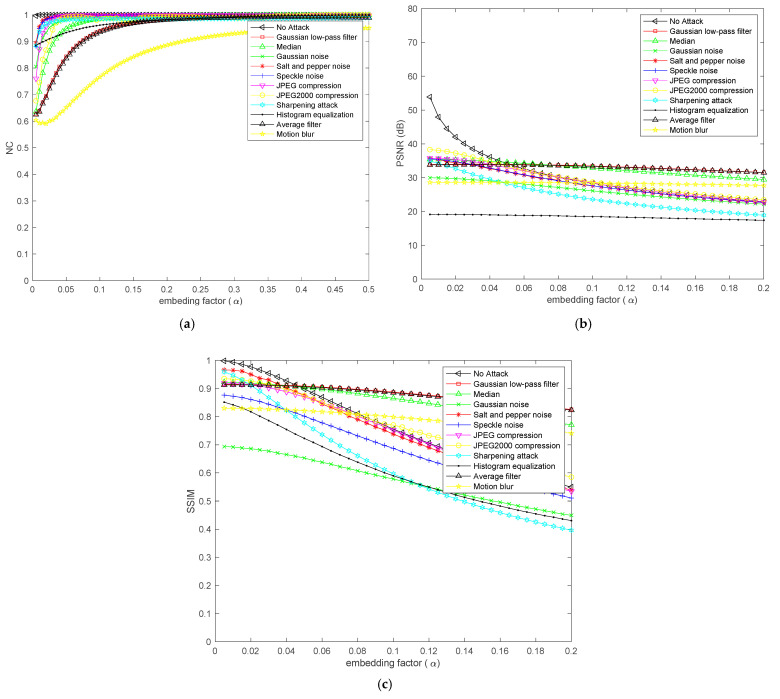
(**a**) Change curve of NC (**b**) Change curve of PSNR (**c**) Change curve of SSIM.

**Figure 9 entropy-24-00246-f009:**
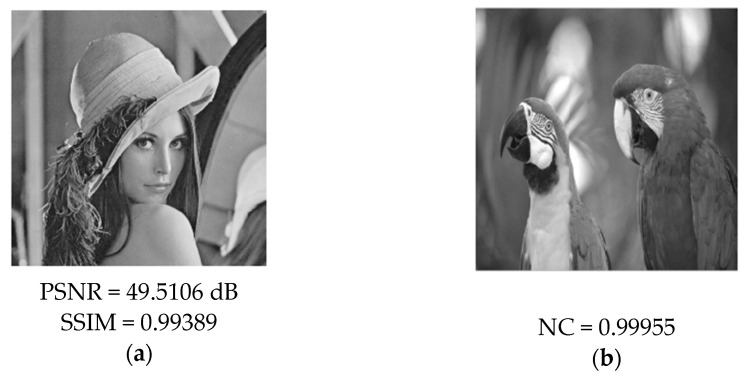
(**a**) Steganographic image and Value of PSNR, SSIM (**b**) Extracted secret image and Value of NC.

**Figure 10 entropy-24-00246-f010:**
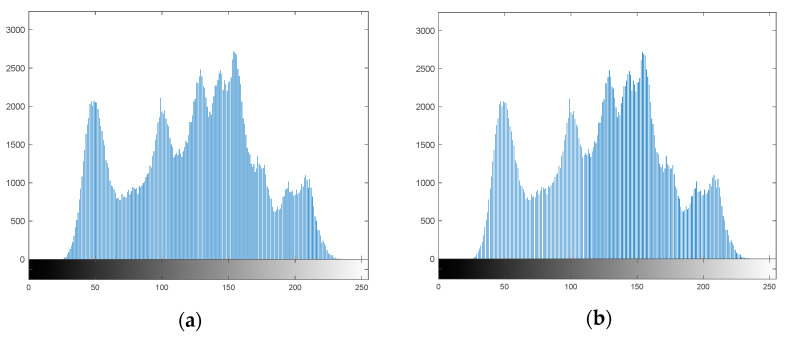
(**a**) Cover image histogram (**b**) Steganographic image histogram.

**Figure 11 entropy-24-00246-f011:**
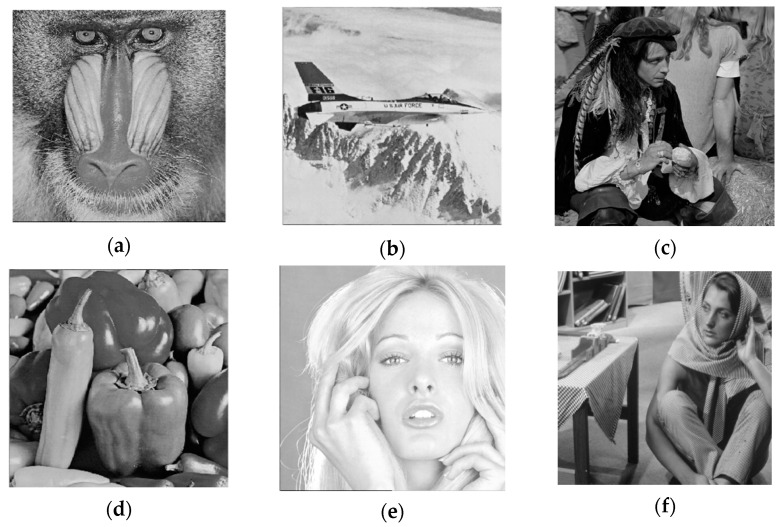
Steganographic images. (**a**) Steganographic image Baboon; (**b**) Steganographic image Airplane; (**c**) Steganographic image Man; (**d**) Steganographic image Peppers; (**e**) Steganographic image Tiffany; (**f**) Steganographic image Barbara.

**Figure 12 entropy-24-00246-f012:**
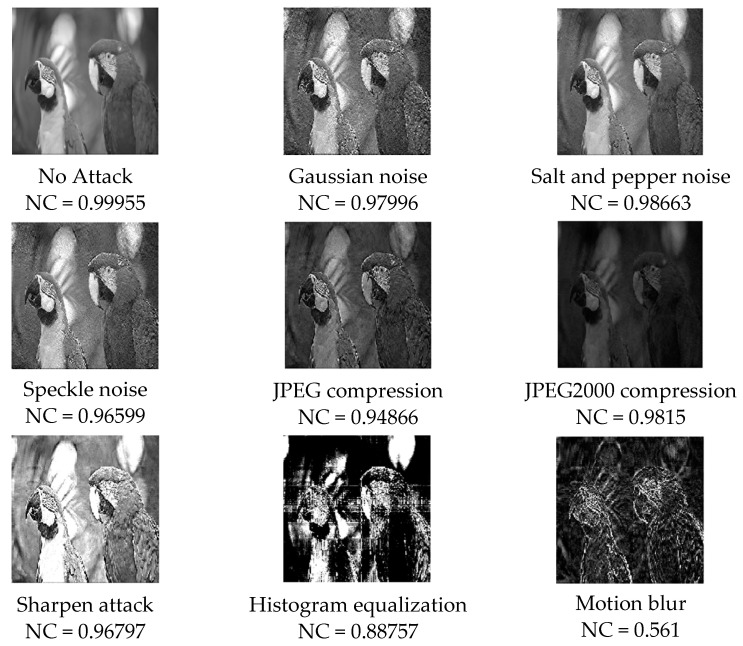
The secret image extracted from the attacked steganography image.

**Figure 13 entropy-24-00246-f013:**
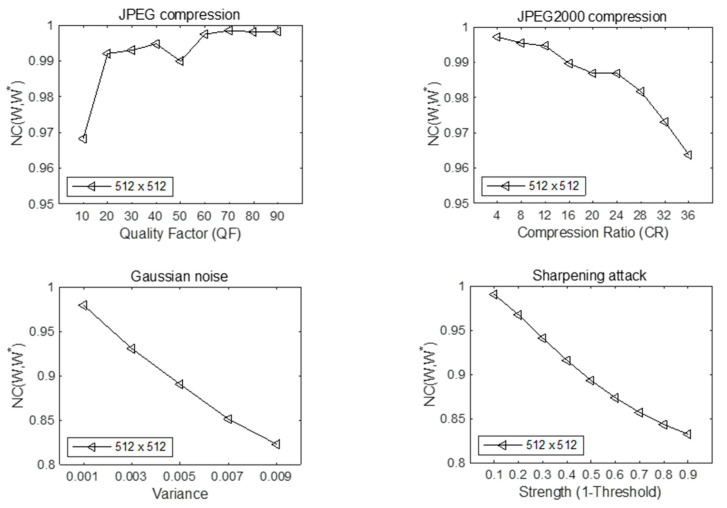
Normalized coefficient values under different attack parameters.

**Table 1 entropy-24-00246-t001:** Attack types and parameters.

Gaussian Noise	Salt and Pepper Noise	Speckle Noise	Compression Attack	Histogram Equalization	Motion Blur	Sharpen Attack
0.001	0.001	0.001	JEPG	—	—	Sharpen 0.2
			JEPG2000			

**Table 2 entropy-24-00246-t002:** PSNR value, SSIM value and NC value under different cover images.

Steganographic Image	Embedding Factor α	PSNR	SSIM	NC
Lena	0.01	49.5106 dB	0.99389	0.99955
Baboon	0.01	49.488 dB	0.99792	0.99966
Airplane	0.02	49.1286 dB	0.99494	0.99897
Man	0.01	57.448 dB	0.99938	0.99376
Peppers	0.01	49.525 dB	0.99792	0.99944
Tiffany	0.02	49.3406 dB	0.99317	0.99961
Barbara	0.01	49.668 dB	0.99637	0.99938

**Table 3 entropy-24-00246-t003:** Comparison with traditional DWT-SVD scheme.

Scheme	Cover Image	PSNR	SSIM
Proposed Scheme	Lena	49.5106 dB	0.99389
	Baboon	49.488 dB	0.99792
	Peppers	49.525 dB	0.99787
Traditional DWT-SVD	Lena	34.0904 dB	0.99566
	Baboon	34.1014 dB	0.99879
	Pepper	34.1188 dB	0.99738

**Table 4 entropy-24-00246-t004:** Concealment comparison of algorithms.

Algorithm	Size of Secret	PSNR	SSIM	NC
R. Thabit et al. [44].	49,152 bits	43.29 dB	—	—
Sajasi et al. [45]	256 × 256	47.78 dB	—	—
Kanan et al. [46]	256 × 256	45.12 dB	—	—
Gulave et al. [47]	78.7 KB	39.84 dB	0.953	—
Subhedar et al. [48]	256 × 256	49.0369 dB	0.9963	—
Proposed algorithm	256 × 256 70 KB	49.5106 dB	0.99389	0.99955

**Table 5 entropy-24-00246-t005:** NC value of extracted secret image.

Attack Type	Baboon		Pepper		Lena	
	NC		NC		NC	
	Two-Level Matrix Decomposition	Single Singular-Value Decomposition	Two-Level Matrix Decomposition	Single Singular-Value Decomposition	Two-Level Matrix Decomposition	Single Singular-Value Decomposition
Gaussian Nosie	0.98968	0.921	0.97797	0.97565	0.97996	0.95773
Salt and pepper noise	0.98997	0.96221	0.98466	0.9469	0.98663	0.93113
Speckle noise	0.98315	0.96367	0.96114	0.94123	0.96599	0.94122
JPEG compression	0.95727	0.94321	0.94917	0.92172	0.94866	0.89297
JPEG2000 compression	0.88304	0.86265	0.98098	0.96023	0.9815	0.88772
Sharpening attack	0.91732	0.89983	0.9732	0.92113	0.96797	0.93211
Histogram equalization	0.85805	0.83122	0.9384	0.91111	0.88757	0.86877
Motion blur	0.51954	0.48999	0.50069	0.4902	0.561	0.49222

## Data Availability

Data is contained within the article.

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
