# Peer review of "Double-Matrix Decomposition Image Steganography Scheme Based on Wavelet Transform with Multi-Region Coverage"

_entropy, 2022, doi:10.3390/e24020246_

Round 1
Reviewer 1 Report
Comments on the Article “Double matrix decomposition image steganography scheme based on wavelet transform with multi-region coverage”
On the basis of ensuring the quality and concealment of steganographic images, this paper pro-poses a double matrix decomposition image steganography scheme with multi-region coverage to solve the problem of poor extraction ability of steganographic images under attack or interference. First of all, the cover image is transformed by multi-wavelet transform, and the hidden region covering multiple wavelet subbands is selected in the wavelet domain of the cover image to em-bed the secret information. After determining the hidden region, the hidden region is processed by Arnold transform, Hessenberg decomposition and singular value decomposition. Finally, the secret information is embedded into the cover image by embedding intensity factor. In order to ensure the robustness, the hidden region selected in the wavelet domain is used as the input of Hessenberg matrix decomposition, and the robustness of the algorithm is further enhanced by Hessnberg matrix decomposition and singular value decomposition. Experimental results show that the proposed method has excellent performance in concealment and quality of extracted se-cret images, and secret information is extracted from steganographic images attacked by various image processing attacks, which proves that the proposed method has good anti-attack ability under different attacks.
This work is relevant to my field and acceptable after major revision with the comments
- Remove the grammatical errors and typo mistakes throughout the manuscript.
- Literature survey is not fine. However, the most recent articles may please be referenced.
- Why this method “steganography scheme based on wavelet transform” is suitable? Kindly discuss in the manuscript.
- Can this scheme is more suitable for colored picture?
- Some terms are not defined properly in the manuscript
- Discuss the term “strength factor” properly in manuscript
- What is intensity factor? Who it will effect?
- What is Logistic chaotic map?
- Explain the major difference between Hessenberg decomposition and singular value decomposition.
Reviewer 2 Report
- several language errors, please use spelling/grammar checker. Examples:
- line 27: at
- line 34 : very long sentence:
- line 42 : very long sentence
- line 115: genetic algorithm, the transformation.... --> missing a connector
- line 550 : consider splitting into two sentences.
- consider breaking the paragraph starting at line 42 into 2 paragraphs
- consider moving the paragraph starting at line 84 after discussing "spatial domain" techniques (line 70). I believe the transition to "transform domain techniques" will be smoother and more intact. the same applied to the paragraph at 94.
- generally, the "introduction" section needs to be carefully revised and restructured.
- in section 2.1, despite the detailed information describing wavelet transforms, the authors didn't cite any reference. the same applies to sections 2.2, 2.3, 2.4, and 4.2.
- figure 1, is poor in quality. All subbands appear black. the same is true for figure 2(b)
- the first paragraph in section 3.1 presents redundant information already discussed in section 1. the same is true for section 4.3
- The " logistics chaos system" used to generate the secret key is explained so late in the method. Consider moving this to section 2.
- step 8 in algorithm 2 needs attention.
- in the embedding algorithm the authors mentioned in line 317 : " The size of the cover image is M×M, the size of the secret image is M×M, and the size of the output image with secret information is M×M." However, the experimental results use cover and secret images of different sizes. I think this needs some explanation.
- the need and use of the " embedding strength factor " were not discussed in the method. It is actually mentioned for the first time in the experimental results.
- parts of figure 8 are repeated. consider resizing as the detail of the curves is not clear.
- line 367 : the terms PSNR, SSIM, and NC are used before being properly defined.
- line 465: the authors mentioned the "traditional singular value embedding algorithm" without any reference to the publication.
- Table 2, is missing some information about the " embedding strength factor ". I guess it should be different for each image used.
- the text in section 4.4 doesn't mention the image used to generate the results in this set of experiments. is it Lena?
- line 538: wrong section number. It should be 4.5.
- "Computational complexity" is not usually measured using execution time. use asymptotic measures such as big-O.
- line 542: the authors mentioned: " Compared with the existing steganography schemes". however, no results are included to support their claim.
- the authors didn't discuss why their method is suitable only for grayscale images and if it is possible to generalize to colored images or not.
Reviewer 3 Report
This article presents a double matrix decomposition image steganography scheme based on widely known methods, such as : wavelet analysis, Arnold transform, Hessenberg decomposition and singular value decomposition.
Furthermore, the manuscript is similar to the following paper:
Junxiu Liu, et. Al. “An Optimized Image Watermarking Method Based on HD and SVD in DWT Domain”, IEEE Access, Volume 7, 2019.
https://ieeexplore.ieee.org/document/8709684
Therefore, the originality of the manuscript must be explained in detail to be accepted.
Regarding the evaluation of the result, I recommend incorporating a discussion about the type of mother wavelet, improve the pseudo-code format of the proposed algorithms and add references in section2 (2.1 to 2.4).
Round 2
Reviewer 1 Report
Now, it's ok. Accept.
Author Response
Dear Reviewer:
Thank you for your comments concerning our manuscript entitled “Double matrix decomposition image steganography scheme based on wavelet transform with multi-region coverage” (ID: Entropy- 1545810). Your comments are very valuable and constructive to the improvement of our paper. At the same time, thank you for your affirmation of our reply and revision of the paper. We hope will be approved to allow successful acceptance of our manuscript for publication.
Thank you again for your help and hope to learn more from you!
Sincerely,
Ping Pan

Reviewer 2 Report
regarding my previous comment:
Point 15: "Computational complexity" is not usually measured using execution time. use asymptotic measures such as big-O.
The authors responded with some modifications in section 4.5. However, the
time complexity analysis provided is not correct. No way this complicated algorithm, with that many steps, has O(1) complexity.
My recommendation is to remove this part from the results section as it doesn't add any real value to the paper.
Reviewer 3 Report
Tthe authors have answered my suggestions
Author Response

(The authors gave the same response as above.)
